# Structural and Physiological Modeling (SAPM) for the Analysis of Functional MRI Data Applied to a Study of Human Nociceptive Processing

**DOI:** 10.3390/brainsci13111568

**Published:** 2023-11-08

**Authors:** Patrick W. Stroman, Maya Umraw, Brieana Keast, Hannan Algitami, Shima Hassanpour, Jessica Merletti

**Affiliations:** 1Centre for Neuroscience Studies, Queen’s University, Kingston, ON K7L 3N6, Canada; maya.umraw@queensu.ca (M.U.); 18bkk1@queensu.ca (B.K.); hannan.algitami@queensu.ca (H.A.); shima.hassanpour@queensu.ca (S.H.); 18jsm10@queensu.ca (J.M.); 2Department of Biomedical and Molecular Sciences, Queen’s University, Kingston, ON K7L 3N6, Canada; 3Department of Physics, Queen’s University, Kingston, ON K7L 3N6, Canada

**Keywords:** functional magnetic resonance imaging, pain, nociception, neural signaling, human, methods, connectivity, brainstem, spinal cord, structural equation modeling

## Abstract

A novel method has been developed for analyzing connectivity between regions based on functional magnetic resonance imaging (fMRI) data. This method, termed structural and physiological modeling (SAPM), combines information about blood oxygenation-level dependent (BOLD) responses, anatomy, and physiology to model coordinated signaling across networks of regions, including input and output signaling from each region and whether signaling is predominantly inhibitory or excitatory. The present study builds on a prior proof-of-concept demonstration of the SAPM method by providing evidence for the choice of network model and anatomical sub-regions, demonstrating the reproducibility of the results and identifying statistical thresholds needed to infer significance. The method is further validated by applying it to investigate human nociceptive processing in the brainstem and spinal cord and comparing the results to the known neuroanatomy, including anatomical regions and inhibitory and excitatory signaling. The results of this analysis demonstrate that it is possible to obtain reliable information about input and output signaling from anatomical regions and to identify whether this signaling has predominantly inhibitory or excitatory effects. SAPM provides much more detailed information about neuroanatomy than was previously possible based on fMRI data.

## 1. Introduction

Functional magnetic resonance imaging (fMRI) data may contain a wealth of information that is unused with current standard analysis approaches. There are limitations of fMRI, such as the lack of ability to distinguish inhibitory and excitatory signaling, that may be mitigated by exploiting more of the information contained within the data. We have previously demonstrated the concept of structural and physiological modeling (SAPM) as a means of fMRI data analysis and its application when used to investigate individual differences in nociceptive processing in the human brainstem and spinal cord [1]. This method has been shown to provide detailed information about neural signaling across networks of interconnected regions, including information about whether each input signal to a region has predominantly an inhibitory or excitatory effect. The objective of the present study is to further validate the SAPM method, provide evidence of reliability and reproducibility, and demonstrate practical aspects of how it can be applied to study human pain processing in the brainstem and spinal cord.

The data used to test the SAPM method were obtained in previous studies, as in our prior proof-of-concept study [1]. The steps undertaken in the present study include (1) validating the network model, (2) determining statistical thresholds needed to infer significance and accounting for multiple comparisons, (3) identifying sub-divisions of regions that best fit the fMRI data, and (4) demonstrating the reliability of the results. Evidence of reliability is based on showing consistency and reproducibility across participants/studies, variations related to pain behavior, and the dependence of results on the choice of network model. We further test the reliability by investigating the correspondence between inferred excitatory and inhibitory signaling and the known neuroanatomy and neurophysiology. The outcomes demonstrate the reliability and validity of SAPM as a means of analyzing fMRI data and increasing the amount of information that can be obtained.

## 2. Materials and Methods

Data were used from three previous studies with similar methods and stimulation paradigms, with some notable exceptions. These data were obtained in previous studies carried out between May 2015 and February 2021 [2,3]. All studies were reviewed and approved by our institutional research ethics board. The studies investigated (a) differences in responses between runs with a noxious stimulus (Stim) and those with no stimulus at all (No Stim); (b) differences in responses between runs with a noxious stimulus (Pain) and runs with an innocuous warm stimulus (Sensation); and (c) differences in responses between runs when participants were told they would feel a calibrated noxious heat stimulus (High) and those when they were told they would feel a less painful heat stimulus which was set at 1 °C lower (Low), even though the same stimulus was applied in every run. Each of the studies involved a paradigm in which participants were told one minute before the stimulation period which stimulus to expect, with thirty seconds of stimulation (or no stimulus) followed by two minutes of “baseline”. The paradigms thus involved periods when the participants could anticipate the stimulus, experience the stimulus, and know that the stimulus had finished. This method enables investigations of emotional and cognitive influences on pain responses, as well as investigations of the effects of different noxious stimuli. The conditions referred to as “Stim”, “Pain”, and “High” are also very similar and have been pooled together in previous studies [1].

### 2.1. Participants

Participants had no previous history of serious neurological injury or serious illness which could have affected their pain responses or fMRI data. Each study is described in detail in previously published papers. The three studies involved a total of 57 participants, consisting of 33 females and 24 males. In all but two cases, data were obtained from all participants in test and control conditions, allowing for within-subjects comparisons of effects. The participant characteristics, including the pain ratings in each condition, are summarized in Table 1.

### 2.2. Participant Training and fMRI Study Methods

The study procedures have been described previously and are summarized here. Each participant underwent an initial 1-h training session followed by an imaging session including fMRI data collection. During training, participants were introduced to the experimental pain stimulus and study design. They were also trained on how to rate their pain using a standardized numerical pain intensity scale (NPS) ranging from 0 to 100 [2,3,4].

The stimulus used for each study condition consisted of repeated brief heat contacts to produce a C-fibre-mediated pain response with temporal summation of second pain [5]. Previous studies have also demonstrated that this method produces a robust BOLD response. The stimulus was applied to the skin overlying the thenar eminence (base of the thumb) on the right hand. In earlier studies, it was applied with an MRI-compatible Peltier thermode (Medoc, Ramat Yishai, Israel) and, in more recent studies, this was replaced with a custom-made robotic contact-heat thermal stimulator (RTS-2). The Peltier thermode was held at a constant temperature and stimuli were applied in a series of brief contacts under manual control by an experimenter who was cued by auditory prompts to ensure consistent timing. The RTS-2 device applied contact heat stimuli repeatedly at the desired timing and temperature under software control (MATLAB, version R2021b, Natick, MA, United States).

Participant training included a series of calibration tests with varying stimulus temperatures (between 40 °C and 52 °C). This procedure familiarized the participant with the stimulus, the study procedures, and the pain rating scale and was used to calibrate the temperature to elicit a pain rating of 50 ± 10 NPS units. Participants also experienced a practice fMRI run in a mock-up of the MRI system to add to their familiarity. They were asked to practice mentally rating each heat contact and to remember their rating for the last contact in the set.

After the training session, the imaging session included 10 trials of our “threat/safety” paradigm [3]. Runs with different types of stimuli (such as “Stim” and “No Stim”, respectively) were randomly interleaved across the 10 runs (5 of each type). Details of the stimulation paradigms are provided in Table 1 and Figure 1. Participants were reminded to mentally rate each heat contact and to report the rating for the last contact when prompted at the end of the run. Each run consisted of 1 min during which the participants did not know what to expect. At the 1-min mark, they were informed via a rear-projection display which type of stimulus would be applied, depending on the study. At the end of each run with stimulation, participants were asked to report their pain rating for the last contact in the run. If a run did not include a stimulus, the run consisted of the same total of 4.5 min of scanning with no stimulus applied.

### 2.3. Functional MRI Data Acquisition

Functional MRI data were acquired at 3 tesla in a Siemens Magnetom Trio until November 2019 and then in a Siemens Prisma system following an upgrade to the system electronics. Test scans were carried out pre- and post-upgrade to confirm that there were no significant changes in image quality, signal-to-noise ratio, or BOLD sensitivity as a result of the upgrade. The fMRI data were acquired using our established method with a T_2_-weighted half-fourier single-shot fast spin-echo sequence [6]. Images spanned a 3D volume extending from the first thoracic vertebra to above the thalamus in nine contiguous sagittal slices. Data were acquired with a repetition time (TR) of 6.75 s/volume; an echo time of 76 msec, to optimize the T_2_-weighted BOLD sensitivity; and a 28 × 21 cm field-of-view with a 1.5 × 1.5 × 2 mm^3^ resolution. Our method employed multiple short fMRI runs, which were combined to provide good statistical power without the disadvantages of long runs (increased chance of motion, participant sensitization or adaptation, etc.). A total of 200 volumes (over 5 repeated runs) were acquired for each participant. This approach enabled participants to verbally report their pain ratings for each stimulation period and to have a brief rest between each run.

### 2.4. Analysis Methods

#### 2.4.1. Pre-Processing

Functional MRI data were pre-processed using analysis software developed in our lab for all levels of the central nervous system [1]. This software, “Pantheon”, is freely available on GitHub (https://github.com/stromanp/pantheon-fMRI (accessed on 7 November 2023). Pre-processing included conversion from the DICOM to the NIfTI format, co-registration using 3D non-rigid-body mapping to correct for bulk body movement, slice timing correction, interpolation to 1 mm^3^ resolution, and spatial normalization to a standardized template spanning the brainstem and spinal cord. The anatomical reference image (template) was defined spanning across the brain and spinal cord regions by combining the MNI152 template and PAM50 template, as described by De Leener et al. [7]. Corresponding anatomical region-of-interest maps were defined from multiple sources, including the CONN15e software package, freely shared anatomical maps, and anatomical descriptions [8,9,10,11,12,13,14,15,16,17,18,19]. These sources were combined to create a single anatomical map. Physiological noise was modeled based on peripheral pulse recordings, bulk motion, and global signal variations in white matter and was then fit to the data and subtracted to remove this component of noise. The first two volumes of each run were omitted from the analysis to avoid periods without consistent T_1_-weighting. These were replaced with copies of the third volume for the purposes of displaying time-series responses. The time-series responses for each voxel were then converted to a percent signal change from the time-series average for subsequent analyses.

#### 2.4.2. Anatomical Regions

The anatomical regions to be used for subsequent analyses were identified using the region maps described above. Entire anatomical regions are not expected to be uniformly involved in nociceptive processing and the regions were, therefore, each divided into 5 sub-regions of approximately equal volumes. Voxels within each anatomical region were grouped into sub-regions using a k-means clustering method applied to the time-series fMRI data concatenated across runs in all participants. As a result, the grouping of the voxels into sub-regions was based on the similarity of time-series responses across the entire study group. The choice of 5 sub-regions was found in previous studies to provide a balance of sufficiently large sub-region volumes and to be sufficiently small for anatomical distinctions to exist between sub-regions [2,3,19]. The regions include the right dorsal region of the 6th cervical spinal cord segment (C6RD), dorsal reticular nucleus of the medulla (DRt), hypothalamus, locus coeruleus (LC), nucleus gigantocellularis (NGc), nucleus raphe magnus (NRM), nucleus tractus solitarius (NTS), periaqueductal gray (PAG) region, parabrachial nuclei (PBN, medial and lateral divisions), and medial thalamus. The same sub-region definitions were used for all analyses for consistency.

### 2.5. Validating the Network Model

#### 2.5.1. Network Analysis

The connections between regions of the brainstem and spinal cord that are involved with nociceptive processing have been described in detail by Millan [10] and are summarized in Figure 2. However, much of the current understanding of the regions involved with nociceptive processing and the connections between them are based on behavioral studies and investigations in animal models and our understanding may be incomplete. It is desirable to create a detailed model in an effort to be realistic and provide as much information as possible. However, it is also necessary to keep the number of fit parameters as small as possible to avoid over-fitting the data. We, therefore, aimed to identify which regions and connections should be included in the model based on the regions with consistent relationships between time-course responses across data sets. This was achieved by applying structural equation modeling (SEM) [1,20], with each region modeled as the target (receiving incoming neural signaling) and two other regions selected as the sources of the incoming signaling. That is, with S_target_ representing the time-series response of the target region and S_source1_ and S_source2_ representing the time-series responses of the source regions then $S_{target}=\beta_{1}S_{source1}+\beta_{2}S_{source2}$. The weighting factors β_1_ and β_2_ reflect the influence of each source on the target and, thus, reflect the strength of the connectivity. Note that this “two-source” SEM method was used only for validating the network model; whereas, the full network model was used for the SAPM method. This process was repeated with all possible combinations of source and target regions in the data set, using data from one participant at a time. The results were, therefore, not constrained to match known or suspected connections between regions but rather show all apparent relationships between all regions. The results were analyzed to identify connections with average β values that are significantly different than zero (indicating consistent connections) or that are correlated with the pain ratings obtained at the end of each run (indicating physiologically relevant variations).

#### 2.5.2. Identifying Connections

The results of this analysis are summarized in Figure 3 for both the significant average connections and the connections with β values that are correlated with pain ratings. Significance was inferred at a family-wise error rate (Bonferroni) corrected p_fwe_ < 0.05, accounting for the total number of possible network combinations that were tested across the anatomical sub-regions. This analysis identified a number of apparent connections that were not included in the description by Millan [10]. However, given that only two sources were used to model each set of possible connections for this analysis, relationships between regions may appear to occur even if they are mediated through a third region. The direction of the influence from one region to another may also not be specific. However, in the few cases in which the influence appears to be only in one direction (such as NGc → C6RD), it is inferred that this relationship is stronger and/or more consistent across participants than the reverse influence (i.e., C6RD → NGc). It is also possible that there exist connections that were not identified by this analysis because the effect is highly variable across individuals or did not contribute significantly with the mode of stimulation that was employed. Nonetheless, this analysis has identified a number of consistent relationships between regions. Some notable relationships that were not included in our initial model (Figure 2) are between the hypothalamus and PBN, the NTS and DRt, and the NRM and NGc. It is equally important that some connections that were expected to occur based on previous studies were not detected in the results, such as NTS → C6RD, PAG → NGc, hypothalamus → LC, and others. We, therefore, selected our network model for SAPM analyses based on these findings and further investigated the plausibility of additional connections that were not included in the original model (Figure 2).

The paper by Millan [10] is focused on descending pain modulation pathways and does not mention PBN–hypothalamus connections. Signaling between the PBN and hypothalamus has been described extensively, however, in the context of ascending signaling related to emotional and autonomic influences on pain [21,22,23].

Prior studies have demonstrated reciprocal connections between the DRt and the spinal cord dorsal horn (DH) and also connections between the NTS and DH [24,25,26,27]. Both the NTS and DRt project signaling to the DH and play important roles in descending pain regulation; ascending projections to the DRt also have been shown to be important. However, tracer studies in rats do not appear to show evidence of direct anatomical connections between the DRt and NTS. It is, therefore, likely that the apparent relationships between BOLD responses in the DRt and NTS that were observed with 2-source SEM are the result of these two regions having strong connections to the spinal cord DH. Similarly, it is expected that the apparent relationships between the PAG and DRt and the NGc and DRt that were identified with 2-source SEM are the result of connections via the DH.

The regions within the rostral ventromedial medulla (RVM), namely, the NRM and NGc, are frequently described together as a single region with multiple functions [28]. However, the nomenclature appears to differ between human and animal studies. In some sources, the NRM is called the RVM and the NGc is referred to as a separate region. A literature search did not identify any published studies with human participants that investigated distinct roles of the NRM and NGc or the possibility of signaling between these regions, except for our prior studies [29,30]. A small number of animal studies have suggested that these regions have distinct functions [31,32]. Based on these studies and the two-source SEM results, we opted to include the possibility of reciprocal NRM–NGc signaling within our network model.

As a result of these analyses, and prior descriptions from published sources, the network model that we adopted matches the depiction in Figure 4. This model also includes variable latent inputs to the LC and NTS and a “fixed” latent input to the spinal cord at C6 (C6RD). The LC was selected because it receives input from cortical regions involved with cognitive and emotional influences on pain and is a source of noradrenaline [10]. In contrast, the NTS is involved with autonomic responses and its input includes ascending input via the vagus nerve. It is, therefore, expected that the input to the network via these regions accounts for several potentially important influences on pain responses. The latent input to the C6RD is modeled based on the timing of the stimulation paradigm convolved with the canonical hemodynamic response function (HRF). This latent input is considered fixed because only its amplitude is varied to fit the model to the data and the temporal profile remains fixed.

### 2.6. Structural and Physiological Modeling (SAPM)

#### 2.6.1. The Basic Concept

SAPM combines *a priori* knowledge of anatomy, neurophysiology, and the relationships between physiological processes and blood oxygenation-level dependent (BOLD) MRI signal variations. This information is used to model the neural signaling underlying observed BOLD responses across interconnected networks of regions. SAPM is an extension of our SEM method because the directions of interactions are inferred based on anatomical information and the fitting method is based on the linear regression between time-series responses. This differs from the original SEM method and later variations that were applied to neuroimaging data, which were based on modeling the covariance between regions [33,34,35]. The BOLD signal for a region in the model is the average over the voxels within a sub-region, as described above. The use of sub-regions (clusters of voxels) addresses the problem that analyzing networks is not plausible on a voxel-by-voxel basis. Moreover, as mentioned earlier, entire regions that are defined based on anatomical features are not expected to be uniformly involved with neural signaling. For example, there may be left/right side differences in function within a region.

The basic concept underlying SAPM is that since the BOLD response is known to relate to pre-synaptic input signaling and the metabolic demand that is driven by incoming neural signaling, the input signaling to each region, S_input_, can be modeled as the sum of the outputs from other regions in the network:$$S_{input}=M_{input}S_{output}$$

Here, M_input_ is a matrix of “D” values, which are the weightings of each incoming signal as they are summed to produce the net total input signaling from each region, and S_output_ is the output signaling from each region. Note that all D values are positive values and allow for the output signaling from a single region to contribute different amounts of input to other regions. Note also that this approach differs from that described previously and is less complex because the previous approach allowed for multiple outputs from each region [1]. The neural signaling, whether as an input or output to/from a region, is represented as the relative drive in metabolic demand. The modeled output signals represent the effect they will have when considered input signals to other regions. All of the modeled signals are thus equivalent to BOLD signals and include the time-lag and smoothing effects of the hemodynamic response. If the incoming signal gets larger (i.e., increased release of neurotransmitters at synapses on dendrites), it will produce an increase in the metabolic demand of the region. The output signaling from each region is modeled similarly to (Figure 5):$$S_{output}=M_{output}S_{output}$$

Here, M_output_ is a matrix of weighting factors that are related to how each input signal influences the output signaling from the region. A positive weighting factor corresponds with excitatory input (more input produces more output); whereas, a negative value corresponds with inhibitory input (more input produces less output). The weighting factors in M_output_ are termed “DB” values because they are the product of the D values mentioned above and the B values that reflect how the incoming signal is converted to contribute to the output signal within each region. As shown in the example in Figure 5 (and the more general version in Figure 6), the matrix of D or DB values contains non-zero values where there is a connection between regions or zero where there is no connection modeled between regions. The matrices M_input_ and M_output_ are very similar in structure, with non-zero values at the same locations. There is, therefore, a value of D for every value of DB; the values of B can be determined by dividing DB by D. The choice of network model, consisting of which connections to include and which not to include, determines the locations in these matrices where there are non-zero values. The B and DB values demonstrate the “apparent transmission effect” between regions, as described previously [1].

One additional key component of the model is the source of the signal variations. The regions in the network cannot drive the signal variations in time themselves; therefore, the network requires unknown (i.e., “latent”) input signaling from outside of the modeled network. The matrix M_output_, therefore, has more rows than M_input_, with the additional row(s) representing the latent input(s). The M_input_ matrix is rectangular, with the number of rows being equal to the number of regions (Nregion) and the number of columns being N, which is the total number of regions plus latents (the matrix size is, thus, Nregion × N). The M_output_ matrix has a size of N × N. These latent inputs can represent neural signaling arising from sources, such as peripheral stimulation, cognitive or emotional influences from higher brain regions, or efferent and afferent autonomic signals.

#### 2.6.2. Setting Up the Model

The expected values of S_input_ are known because these correspond with the observed BOLD signal variations in each region. The values of D, DB, and S_output_ are not known. However, these values can be estimated using the pre-defined network model to match the modeled S_input_ values to the observed BOLD responses. The method for determining D, DB, and S_output_ is guided by the properties of the matrices in the equations above. The equation $S_{output}=M_{output}S_{output}$ corresponds with the general eigenvalue equation λx = Mx. For any square matrix (as in the case of M_output_) with a size of N × N, it is possible to find eigenvectors (x, of size N × 1) and corresponding eigenvalues (λ, which are scalars, i.e., single values), which make the eigenvalue equation true. For a matrix of size N × N, there will be up to N eigenvectors and corresponding eigenvalues. The eigenvectors and eigenvalues can be thought of as properties of the matrix M. Therefore, for a set of DB values in the matrix M_output_, we can determine the eigenvectors and the corresponding eigenvalues. We need an eigenvector with a corresponding eigenvalue, λ, that is equal to 1. Then, x = Mx, corresponding with $S_{output}=M_{output}S_{output}$. The form of the matrix M_output_, as shown in the example in Figure 5, will always have a portion in the lower right corner of the matrix that matches the identity matrix (i.e., 1′s on the diagonal and 0′s off-diagonal) because of the rows corresponding to the latent inputs. For example, if there are 3 latent inputs, then, this lower right portion of the matrix will be a 3 × 3 identity matrix. As a result, it will always be possible to find an eigenvector corresponding to each latent input that has an eigenvalue equal to 1. This is a property of matrices of this form.

If S_output_ is an eigenvector of M_output_, which has a corresponding eigenvalue equal to 1, then, the equation $S_{output}=M_{output}S_{output}$ is true. In addition, this equation is also true for any multiple of an eigenvector; therefore, the values in S_output_ can be multiplied by any scalar value and the equation is still true (i.e., $S_{output}A=M_{output}S_{output}A$, where A is any scalar value). This equation can, therefore, represent the output signaling of each region at a single time point (because the eigenvectors are N × 1 if there are N regions). The individual values would show the corresponding output signal from each region. That is, if we set the output value from one region, we could determine the corresponding output signaling from every other region. We can, therefore, calculate the eigenvector corresponding to one latent input and then choose any value for the latent input and determine the corresponding output signaling from every region in the network. In effect, this shows how the latent input would propagate through the network and affect every output.

We can also sum the effects of eigenvectors. That is, if we take two eigenvectors corresponding to two different latent inputs, such as $S_{output1}=M_{output}S_{output1}$ and $S_{output2}=M_{output}S_{output2}$, then:$$S_{output1}+S_{output2}=M_{output}S_{output1}+M_{output}S_{output2}\;S_{output1}+S_{output2}=M_{output}{(S}_{output1}+S_{output2})$$

As a result, we can determine the output from every region for multiple latent inputs.

Again, the description so far only describes the output at a single point in time. For any given latent inputs at a point in time, the strength of the output from each region can be calculated. However, S_output_ can be extended to have a size of N × T, where T is the number of time points in the fMRI time-series data. This is because M_output_ describes the signaling relationships between regions and is modeled as being constant in time; therefore, the corresponding eigenvectors also do not vary in time. That is, the values of the latent inputs can be chosen at each time point and the corresponding outputs from each region can be determined for each time point. This is achieved simply by scaling the eigenvector to have a value of 1 corresponding to the latent input. Then, if we multiply the eigenvector by the desired latent input value, the resulting values match the output from each region. That is:$$S_{output}=M_{eigv}L$$

If there are N regions plus latents, then S_output_ has a size of N × T; if there are nL latent inputs, then M_eigv_ has a size of N × nL; and if we choose values at each time point for each latent input, then L has a size of nL × T. Then, $S_{input}=M_{input}M_{eigv}L$.

Another property of the M_output_ matrix is that since:$$S_{output}=M_{output}S_{output}$$
then:$$S_{output}=M_{output}{M_{output}S}_{output}$$

This means that S_output_ consists of eigenvectors of M_output_ and also of M_output_ M_output_. While the above equations are true for S_output_, they are not necessarily true for all possible values of S. Therefore, it is not necessary that M_output_ is an “idempotent” matrix where $M_{output}M_{output}=M_{output}$.

#### 2.6.3. Solving the Model to Estimate Input and Output Signaling

So far, the problem has been described; however, the method to solve S_output_ and the latent inputs has not. The description above relies on having a set of DB values to create the matrix, M_output_, and D values to create M_input_. Given these values, we would then compute the eigenvectors corresponding to each latent input. Then, we would determine the latent inputs, L, for which the estimated (computed) S_input_ matches the observed BOLD time-series variations in each region. We will now refer to the BOLD time-series variations as S_BOLD_ to distinguish the measured values from the computed values, S_input_.

Since $S_{input}=M_{input}M_{eigv}L$, then, we want $S_{BOLD}=M_{input}M_{eigv}L$. We can calculate L as follows:

To simplify the equation slightly we first set:$$M_{1}=M_{input}M_{eigv}$$
then:$$S_{input}=M_{1}L$$
so, therefore:$$S_{BOLD}=M_{1}L$$
and we can reform the equation to solve for L:$${L={\left({M_{1}}^{T}M_{1}\right)}^{-1}{M_{1}}^{T}S}_{BOLD}$$

(Here, the superscript T indicates the transpose of the matrix and the superscript −1 indicates the matrix inverse).

Then, we can calculate S_input_ and determine how well it agrees with S_BOLD_:$$S_{input}=M_{input}M_{eigv}L$$

The residual error can be computed as the sum of the squares of the differences (SSQD):$${err}_{SSQD}=\sum_{region}\sum_{time}{\left(S_{input}-S_{BOLD}\right)}^{2}$$

(The S symbol represents summation).

There are other options for how to express this error term, such as:$$R_{total}^{2}=\frac{\sum_{region}\sum_{time}{\left(S_{input}-S_{BOLD}\right)}^{2}}{\sum_{region}\sum_{time}{S_{input}}^{2}}$$
or:$$R_{average}^{2}=\frac{1}{N_{regions}}\sum_{region}\frac{\sum_{time}{\left(S_{input}-S_{BOLD}\right)}^{2}}{\sum_{time}{S_{input}}^{2}}$$

(The sum over the time points is obtained using data for one region at a time and, then, the values for each region are summed).

Here, $R_{total}^{2}$ is the proportion of the variance that has been explained across all BOLD time-series responses in all regions; whereas, $R_{average}^{2}$ is the average variance that has been explained when it is computed for each region separately. Differences in these values demonstrate effects such as the fit being good in some regions and poor in others or regions with large signal variations having a larger influence over the fitting procedure than regions with small signal variations. This point leads to the issue discussed below, that of dealing with large differences in variance between different regions.

#### 2.6.4. The Gradient-Descent Method to Determine the Values in M_input_ and M_output_

The gradient-descent method consists of randomly selecting initial values of DB and D, within a reasonable range, and computing the initial error value. Then, each DB and D value is changed by a small amount, one at a time, and the change in the error is determined. The change in error relative to the change in the D or DB value is used to estimate the amount and direction in which the value needs to be changed in order to reduce the error. This is done in small repeated steps and, ideally, the error decreases with each step. If the error does not decrease then the size of the increments is decreased and the process is repeated. The values of D and DB are gradually improved with each step until no further improvements can be found. The result is expected to be the optimal fit.

However, additional information is used to help guide the fit. It is desirable to obtain the best fit with the smallest possible values of D and DB. That is, if the same or nearly the same amount of error can be found using different values, the smaller values of D and DB are preferred. We, therefore, included a “cost” term, added to the error term, which is given lower weighting than the error term but, nonetheless, influences the search for the optimal fit. For the present analysis, this was the sum of the absolute DB values (termed “L1 regularization”) [36]. Thus, values with lower absolute values have lower “cost” and are preferred.

In order to avoid local minima in the search for DB and D values for the present analysis, we started at 15 different random locations (sets of values of DB and D) and carried out the gradient-descent approach for 15 iterations. From this point, the gradient descent was continued from whichever point had the highest R^2^ value for a maximum of 250 additional iterations. After this number of iterations, the rate of improvement on each step was confirmed to be extremely small or negligible.

#### 2.6.5. Dealing with Different Variance across Regions

In real-world fMRI data, it is to be expected that some regions will have larger signal variations than others. Given that the signal variations are represented for clusters of voxels representing sub-regions within each region, it is to be expected that signals will be averaged over different numbers of voxels for different regions, resulting in different amounts of variance. We have attempted to mitigate one aspect of this problem by defining sub-regions that have approximately equal volumes within each region. Also, some regions will naturally have larger BOLD signal variations than other regions. This difference can have a strong influence on the fitting method. This is addressed in the current version of the SAPM method by scaling all of the time-series responses to have the same average variance across regions. While scaling the time-series responses does not have any effect on the DB values, it alters the D values. After the fit parameters (D and DB values and latent inputs) have been determined with the scaled data, the fit parameters for the original (unscaled) data can be obtained by scaling the D values. This is easily verified by comparing the original and fit time-series responses (i.e., S_input_ and S_BOLD_).

#### 2.6.6. Validating the Method and Testing Statistical Thresholds with “Null” Data

The SAPM results discussed here are in the form of D and DB values for each person that is studied. Analyses could similarly be carried out with repeated runs within each participant or with data combined across participants within a group, etc. However, these variations are not discussed here. For the purposes of comparing SAPM results with pain behaviors, we opt to identify connections that have group average D or DB values that are significantly different to zero, indicating consistency, or to identify D or DB values that are correlated with pain ratings or other characteristics of pain responses across the group of participants. The statistical measures are, therefore, in the form of T-values, $T=\bar{DB}/sem(DB)$, or R values given by the correlation between DB values and pain ratings or from the linear regression of DB values with pain ratings. Here, $\bar{DB}$ is the average DB value and sem(DB) is the standard error of the mean across the group. R values are then converted to Z scores by means of Fisher’s Z-transform, $Z={tanh}^{-1}(R)/\sqrt{N-3}$, where N is the number of DB values used in the correlation or regression. The probability (p) that a given T or Z value could have occurred by random chance can then be determined using either a Student’s T-distribution or a normal distribution, respectively. While each SAPM fit of the BOLD signal responses to the network model is a single statistical test, there are multiple D and DB values to be tested for significance. The choice of statistical threshold must, therefore, take into consideration the problem of multiple comparisons and be adjusted accordingly.

In order to establish suitable *p*-value thresholds for inferring significance, we first ran tests to validate the assumption that DB values have average values of zero when “null” data are used for SAPM analysis. By definition, D values are always positive, are typically close to values of 1.0, and are, therefore, not tested for significance. For these tests, “null” data consist of normally-distributed random values in place of actual BOLD time-series response values and, thus, cannot produce any true significant results. For the present analysis, each null data set simulated repeated runs in each person, corresponding to our fMRI acquisition methods. Average DB values were tested by running simulations with 1000 sets of null data.

#### 2.6.7. Sub-Region Search

With the SAPM method, it is not practical to test all possible combinations of sub-regions (i.e., clusters of voxels with similar properties) within regions in the network. For a network consisting of 10 regions, with 5 sub-regions each, there are almost 10 million possible combinations. It is therefore useful to identify sets of sub-regions (one per region) that provide the best fit of the model network to the fMRI time-series data. It is expected that there are multiple combinations of sub-regions for which the network fits the data, possibly demonstrating different aspects of the network function. In order to identify these sets of sub-regions, we employed a gradient-descent approach to search for improvements in the average R^2^ value on each iteration, starting from a random selection of sub-regions. Each sub-region was tested for one region at a time and the sub-region that provided the highest average R^2^ value was selected. The process was carried out for all regions in random order and then was repeated for all regions until no further improvement in the R^2^ value could be found.

Once a set of sub-regions was identified, analyses were carried out using the data in these sub-regions for each participant. Significant connections were identified based on the DB values being significantly different than the average values determined with null simulations or DB values being significantly correlated with pain intensity or pain unpleasantness ratings. As a result, the iterative process used for identifying the sub-regions based on R^2^ values does not contribute to the problem of multiple comparisons when inferring the significance of the results.

### 2.7. Applying the Methods to Data from Pain Studies

In order to test the reliability of the results obtained with SAPM, analyses were carried out with data from each of the 55 participants in the high-pain conditions listed in Table 1. Group summary results were obtained for all participants, as well as for each of the study groups, separately (“Stim”, “High”, and “Pain”, as listed in Table 1). These study groups were selected because they were obtained under similar conditions and are the higher pain rating conditions from each of the three previous studies. The differences between the conditions are primarily the interleaved conditions that were applied in each study and may have influenced the pain responses.

Compared to the previous demonstration of the SAPM method, the current method is less complex because each region is modeled as having a single output; although, this may be scaled by different amounts (via D values) for modeling inputs to other regions. The aim of the previous study was to demonstrate the proof-of-concept of the SAPM method and the effectiveness was demonstrated by investigating individual differences in pain responses across individuals. The previous study employed a different network model as well, with one key difference being the choice of variable latent inputs to the thalamus and NTS compared to the present network model with variable latent inputs to the LC and NTS, as described above.

## 3. Results

### 3.1. Results of Tests with “Null” Data

The results of null tests demonstrated that average DB values can deviate from zero (typically less than one standard deviation across the multiple simulations) and these average values depend on the choice of network model. The input and output signaling that is modeled for each region is a complex balance of all of the connections that are intended to represent the neural signaling within the actual anatomy. However, this model is limited to the average signals for each region that is represented in the network model, and the model is incomplete. As a result, even with “null” data, the modeled DB values can have a tendency to be positive or negative, as shown in Figure 7. In order to identify connectivity values that have a low probability of occurring by random chance (i.e., testing the “null hypothesis”), the average DB values from a group of participants for a given condition were, therefore, compared with the results of null simulations. The resulting T-value distributions were confirmed to correspond with Student’s T-distributions, as shown in Figure 8.

Statistical thresholds required to identify significant connectivity values and account for multiple comparisons were determined by simulating 5000 sets of data from 20, 40, and 60 participants. Data sets representing groups of participants were selected at random from the 1000 simulations in order to determine the distribution of T-values. The results indicate that a false-positive error rate of 5% for a single connection in the network can be obtained by applying a Bonferroni correction for the number of connections in the network. That is, for 32 connections in the network (not counting latent inputs), an uncorrected *p*-value of 0.05/32 = 0.00156 is required. This corresponds with a family-wise error-corrected *p*-value of 0.05.

### 3.2. Results of Applying the Methods to Data from Pain Studies

The SAPM results demonstrate consistent connectivity patterns across the three data sets that were analyzed, as shown by the results for each condition and by the combined results across all 55 participants (Figure 9, Table 2). The R^2^ values for each data set and set of sub-regions were consistently larger than the R^2^ values obtained with null data (Table 3). The results vary depending on the sub-regions that were selected in each region, as expected, but the key features reveal similar relationships between regions. The well-described descending pain regulation pathway of PAG → NRM → cord is demonstrated in sub-region set 1, with apparently inhibitory signaling from the thalamus to the PAG, excitatory signaling from the PAG to the NRM, and then inhibitory signaling again from the NRM to the C6RD region of the cord. As shown in the anatomical figures in Figure 10, the C6RD region in this sub-region set is within the deeper dorsal horn laminae. In comparison, sub-region sets 2 and 3 both show ascending inhibitory signaling from the C6RD region to the NRM and, also, ascending excitatory signaling from the C6RD to the NGc, from more superficial laminae near the tip of the dorsal horn. The results also indicate signaling from the LC that is predominantly excitatory to the thalamus and inhibitory to the PBN. There is also an extensive network of connections demonstrated between the NTS, PBN, hypothalamus, and PAG.

Time-course responses are shown in Figure 10 for selected regions of the network for data from all 55 participants. The time courses were determined from three different sets of sub-regions; although, some sets have certain sub-regions in common (such as Thalamus 3 and NRM 2). The sub-region sets were determined using the search algorithm described above to identify the regions with the highest average R^2^ fit to the data. This process was then repeated with different C6RD sub-regions fixed at different values. The values plotted in red are the measured average BOLD responses for the group, corresponding with the total input signaling to each region. Values plotted in blue are the average input and output signals for each region that were computed with SAPM. Periods when participants were informed of the type of stimulus to expect are indicated with blue bars and the periods when the stimulus was applied are indicated with yellow bars. The anatomical distributions of the sub-regions are shown in sagittal and axial slices. The orientations of the anatomical slices are indicated at the bottom of the figure, along with a reference image showing the anatomical distributions of the regions in 3D.

## 4. Discussion

### 4.1. Validation with “Null” Tests

The SAPM analyses carried out with simulated “null” data sets and with data from 55 participants serve to validate the statistical methods and thresholds used to infer significance. The results also demonstrate the reliability of the method based on consistent findings across multiple data sets and the consistency with the known neuroanatomy. Tests with null data have demonstrated that, depending on the choice of network model, DB values can be slightly skewed toward non-zero values. However, the values observed with the null data sets were found to be less than one standard deviation from zero. Nonetheless, we opted to use the null data as a reference in order to identify connections with DB values that had a low probability of occurring as a result of random noise and were, thus, inferred to be significant. Using the null data results as a reference, we confirmed that T-values had distributions that corresponded with the Student’s T-distribution. The Student’s *t*-test is, therefore, valid for inferring the significance of group-average connectivity values. We determined that in order to identify significant connections within a network, the problem of multiple comparisons can be addressed by applying family-wise error rate corrections (the Bonferroni method) based on the number of connections in the network. The results also demonstrate that the average R^2^ values for the fit of the data to the model network had a much lower range in the 1000 simulated null data sets than was observed across all 55 of the participant data sets studied. This suggests that the probability of any one of these participant data sets fitting to the network by random chance is *p* < 0.001 on an individual basis. The results obtained with null simulations confirm that the appropriate conditions are met for the proposed statistical methods to be valid in order to infer the significance of the results obtained with SAPM; they support the conclusion that the measured BOLD responses fit the network model.

### 4.2. Validation with fMRI Data

Results from 55 participants add to our previous proof-of-concept demonstration of the SAPM method. With the current version that is simplified slightly from the original, the results again have consistent features across multiple data sets. As discussed above, the analyses with null data demonstrate that the agreement between the measured and fit data (R^2^ values) could not have occurred by random chance. Results for sub-region set 1, shown in Table 2, demonstrate the consistency across three different sets of data. Although some values for some conditions did not reach statistical significance after correcting for multiple comparisons, the DB values are similar across the studies and are highly consistent in terms of whether they reflect excitatory (positive) or inhibitory (negative) signaling.

The networks shown in Figure 9 demonstrate the PAG → NRM → cord descending modulation pathway with inhibitory signaling from the thalamus to the PAG, excitatory signaling from the PAG to the NRM, and inhibitory signaling from the NRM to a region of the deep dorsal horn in the spinal cord at C6. Other regions of the cord dorsal horn are shown to project ascending inhibitory signaling to the NRM and excitatory signaling to the NGc. The computed time courses of input and output signaling shown in Figure 10 show that in the thalamus, there were consistent signal increases after the participants were informed of the stimulus type and then decreased signals during the period of stimulation. This occurs in both the input and output signaling. Similarly, the input signaling to the PAG showed a gradual increase after the participants were informed of the stimulus type and decreasing input signaling during stimulation. However, the PAG output appeared to be increased when participants were told what to expect and during stimulation. The NRM input signaling has a notable increase from the start of the stimulation paradigm to the time when participants were informed of the stimulus type and decreased input signaling during stimulation. The output signaling from the NRM increased when participants were told what to expect, as well as during stimulation. The signaling in the C6RD region also showed increased input signaling when participants were told what to expect, followed by a decrease in signaling; then, another increase occurred when the stimulus was applied, followed by a decrease in the signal again after the stimulation period. The output signaling from the C6RD depends on the sub-region, as the most superficial sub-region (near the tip of the dorsal horn, C6RD 3) had increased output signaling both when participants were told what to expect and during the stimulation period. In contrast, the deeper dorsal horn sub-regions (C6RD 4 and C6RD 1) had slightly decreased outputs when participants were told what to expect and decreased outputs during the stimulation period.

### 4.3. Comparisons of SAPM Results with Known Neuroanatomy

In order to obtain evidence of the reliability of the SAPM results, we aimed to determine whether the results correspond with the known neurophysiology in terms of inhibitory versus excitatory signaling. The primary neurotransmitters that are involved, and whether they cause inhibition or facilitation of pain, can depend on the type of nociception and how it is influenced. This can be seen in cases such as stress analgesia and emotional or cognitive influences on pain [10].

Pain perception is known to be strongly affected by the administration of opioid agonists to the PAG to produce analgesia or antagonists to facilitate pain and direct opioid signaling is generally inhibitory. While some prior studies describe reciprocal connections between the thalamus and PAG [37], others describe thalamus–PAG pathways in which the effect on pain is mediated by a third region, such as the dorsolateral prefrontal cortex (dlPFC) [38] or the ventrolateral orbital cortex (VLO) [39,40,41,42]. It is possible that the apparent temporal relationships between the BOLD time-course responses in the thalamus and PAG are mediated by a third region that is not included in our SAPM model. The pathway from the nucleus submedius (Sm) of the thalamus to the VLO and then to the PAG is described as activating the brainstem descending inhibitory system [42]. However, it is not clear from published studies whether the net input signaling to the PAG is expected to be inhibitory. Moreover, the sub-regions of the thalamus that were identified by the SAPM analysis are within the medial thalamus and may include the nucleus submedius based on anatomical descriptions [43,44]. The SAPM results for the apparent thalamus→PAG signaling (even if via a third region) are, therefore, plausible but cannot be confirmed with the current data.

In contrast, the signaling pathway from the PAG to the NRM has been studied extensively, involves several different neurotransmitters, and can cause inhibition or facilitation of pain [10]. Nonetheless, an excitatory link from the PAG to the NRM has been described and is consistent with our SAPM results [10]. Signaling from the NRM to the spinal cord dorsal horn is also described as being able to contribute to inhibition or facilitation of pain. The SAPM results indicate that, although the input signaling to the cord dorsal horn is increased during stimulation, the output signaling is increased from superficial regions and decreased from deeper regions. The NRM → C6RD connection that was found to be significant at the group level, after correcting for multiple comparisons, indicates an inhibitory effect of the NRM signaling on the cord output and involves a deeper dorsal horn region. Again, we cannot unequivocally confirm that these results agree with the known neurophysiology, but they are plausible.

The results also indicate signaling from the LC that is predominantly excitatory to the thalamus and inhibitory to the PBN. The LC produces noradrenaline and is part of the reticular activating system and signaling from the LC is expected to be primarily excitatory. It is possible that the apparent inhibitory effect of LC → PBN signaling is mediated via a third region that is outside the modeled network, such as the amygdala [45,46]. The results also show an extensive network of connections between the NTS, PBN, hypothalamus, and PAG that have predominantly excitatory effects. The interconnections across the NTS, PBN, hypothalamus, and PAG have been described extensively in relation to pain regulation and the NTS signaling to the PAG, for example, is expected to be excitatory [47]. The SAPM results, thus, appear to be consistent with the known physiology to the extent that the results from vastly different modalities and conditions can be compared.

The SAPM results thus appear to be consistent with the known neuroanatomy. We have also identified that results may demonstrate connections that are mediated via regions that are not in the network. By including information about the known neuroanatomy in our interpretation of the results, we can account for such possibilities. In addition to the expected features, the present results have demonstrated a number of interesting features about nociceptive signaling that have previously been suggested but have not been demonstrated in humans. For example, the NRM and NGc have distinct functions [1,29]. This is consistent with previous studies and is demonstrated by the different time-course responses and connectivity to different regions. Additionally, results show that different parts of the spinal cord dorsal horn have different functions. While this corresponds with the well-known laminar structure of the spinal cord, the present results show that the connectivity from brainstem regions and the output signaling differ across regions of the dorsal horn. The results also further demonstrate the continuous nature of descending regulation, as shown by the responses during the anticipation of the stimulus, as well as direct responses to the stimulus [3].

### 4.4. Limitations and Future Directions

SAPM is distinct from other methods for computing effective connectivity. SAPM models are less complex than those used in dynamic causal modeling (DCM) [35,48] but are more complex than those used in structural equation modeling (SEM) [20]. SAPM has inherent limitations; however, it also provides potential benefits over other methods. While DCM and SEM model interactions at the nodes of a network, SAPM models interactions, as well as the effects of these interactions, in terms of BOLD signal variations. SAPM, SEM, and DCM differ in the type of information that they provide and in how they can be used to understand neural signaling across a network. A current limitation of the SAPM method is that it relies on the gradient-descent approach to estimate the model parameters and the results can be affected by initial parameter estimates. SAPM also relies on a simple linear model and the results depend on the choice of network model. Output signals and latent inputs may not always have unique solutions; however, our results show that they represent plausible models that fit observed BOLD responses. In comparison, the DCM method has not been used to model the output signaling from regions of a network or to obtain information about whether the input to each region has predominantly inhibitory or excitatory effects. Moreover, we have applied SAPM and SEM, in previous studies [49,50], to data from each participant to investigate individual differences in pain responses. There do not appear to be any examples in the literature of DCM being used in this manner.

A notable difference between SAPM and DCM is that SAPM does not include a hemodynamic model. The modeled input and output signaling is expressed as BOLD responses, or equivalent to BOLD responses, which already include hemodynamic effects. Models of input signaling to the network as a direct result of peripheral stimulation are convolved with the canonical hemodynamic response function [51] before being introduced to the model. As a result, SAPM models BOLD signal variations, in contrast to the neural signaling that is modeled with DCM, and a fixed (canonical) hemodynamic model is assumed.

An important future direction for the development and further validation of SAPM will be to compare it directly with DCM and SEM. This will require the development of simulations of biologically realistic data to enable side-by-side testing of the methods with known connectivity parameters, as has already been demonstrated for DCM [35,48,52]. Comparisons should also include direct comparisons of the results produced by these methods for the same sets of actual fMRI data from participants. Our results have already demonstrated the reproducibility and sensitivity of SAPM and its value for studies of nociceptive processing. Comparisons with other methods are expected to demonstrate the added value provided by the SAPM approach.

## 5. Conclusions

The effectiveness and reliability of the results obtained with SAPM have been demonstrated in multiple ways. We have validated a network model based on the known neuroanatomy and have shown that the fMRI data from each participant fit the network (based on R^2^ values) better than what could occur by random chance as a result of Gaussian noise (*p* < 0.001). The results also validate the statistical methods we have employed for identifying connections with group average values that are significantly different than the values obtained with null data. The SAPM results are shown to have consistent features across different data sets, including where DB values are positive or negative, demonstrating that the network parameters are reproducible. In addition, the time-course responses that are computed for input and output signaling from each region have consistent features across participants and they show sensitivity to variations in emotional, cognitive, and sensory conditions throughout the stimulation paradigm. The properties of connectivity (apparent transmission effect) and output signaling computed with SAPM, including inhibitory and excitatory signaling, have also been shown to be largely consistent with the known neuroanatomy. The results model input and output signaling that can explain observed BOLD signal variations. However, the results depend on the choice of network model and may not include all connections or all regions that contribute to neural signaling. Nonetheless, the results of this study demonstrate that the SAPM method is reliable and effective with an appropriate choice of network model. With SAPM, it is possible to obtain information about input and output signaling from anatomical regions and to identify whether this signaling has predominantly inhibitory or excitatory effects. The additional information obtained with SAPM enables much more detailed interpretations of fMRI results than was previously possible.

## Figures and Tables

**Figure 1 brainsci-13-01568-f001:**
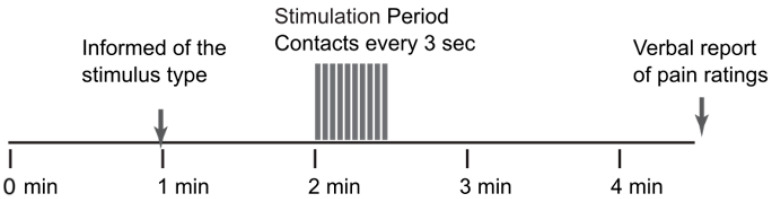
The stimulation paradigm used for each of the fMRI studies.

**Figure 2 brainsci-13-01568-f002:**
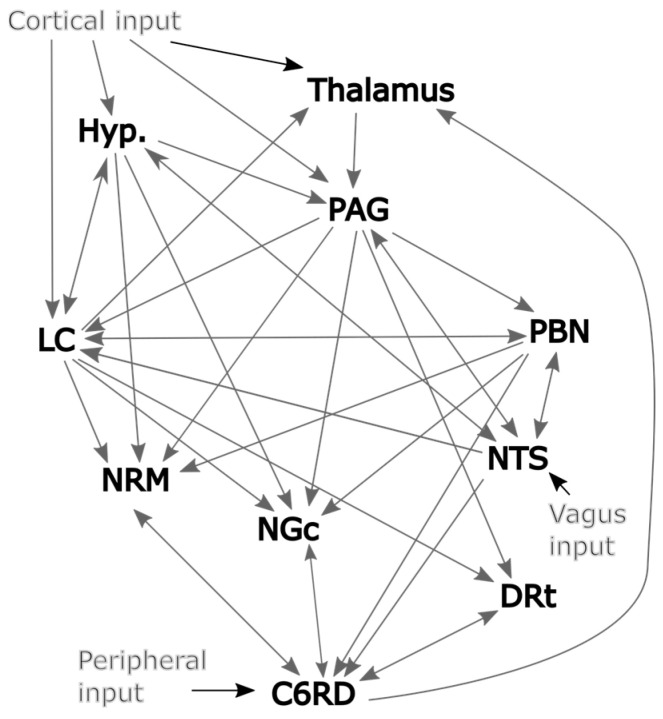
Initial model of plausible network connections based largely on Millan’s work [10].

**Figure 3 brainsci-13-01568-f003:**
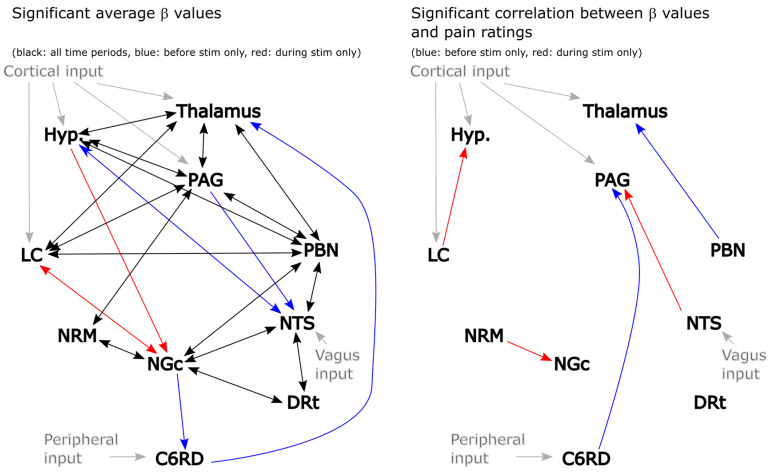
Results of 2-source SEM showing connections with significant average β values (**left**) and connections with β values that are correlated with pain ratings across participants (**right**). Connections between regions are indicated with arrows, with the arrowheads indicating the direction of the influence. Black lines: significant connections both before and during stimulation, red: significant only in the period before stimulation, blue: significant only in the period that included the stimulation period. Gray lines indicate latent (unobserved) inputs from outside the network.

**Figure 4 brainsci-13-01568-f004:**
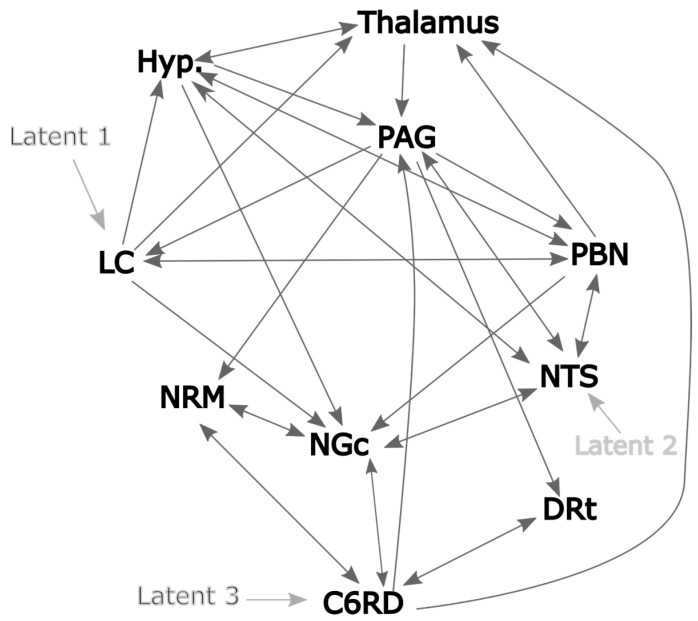
Proposed network model based on 2-source SEM and published sources. The dark gray lines indicate plausible network connections, and the lighter gray lines indicate the chosen latent inputs.

**Figure 5 brainsci-13-01568-f005:**
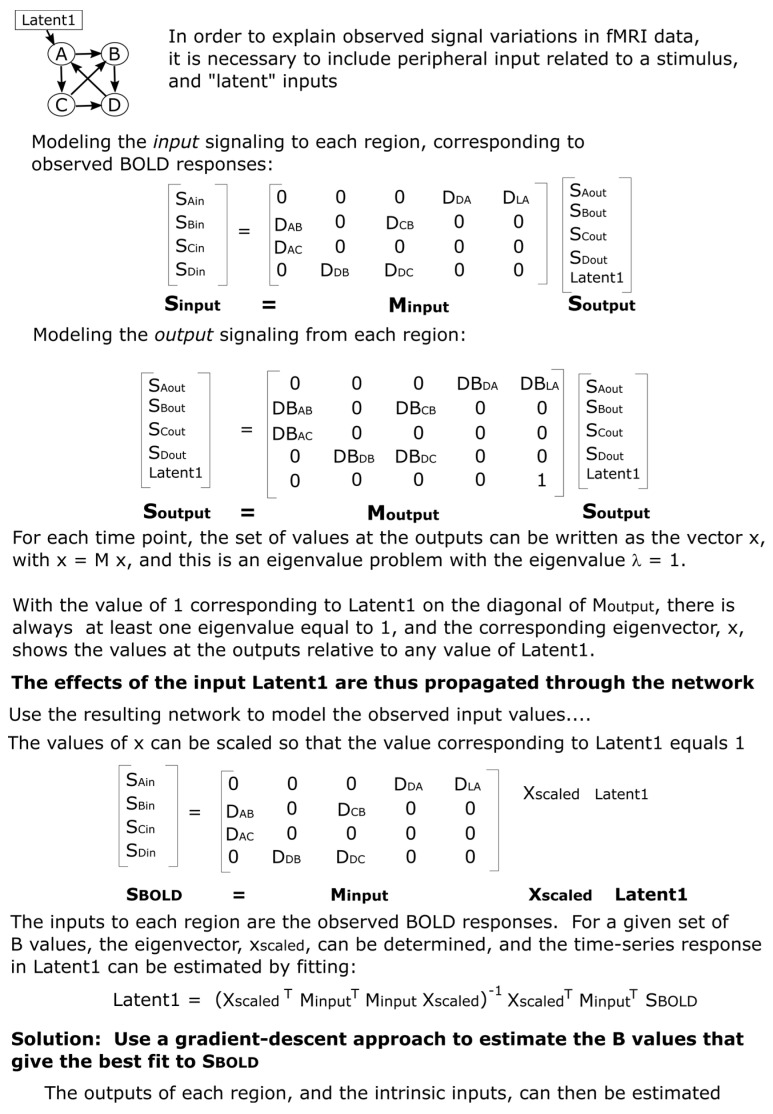
Example illustrating the SAPM method.

**Figure 6 brainsci-13-01568-f006:**
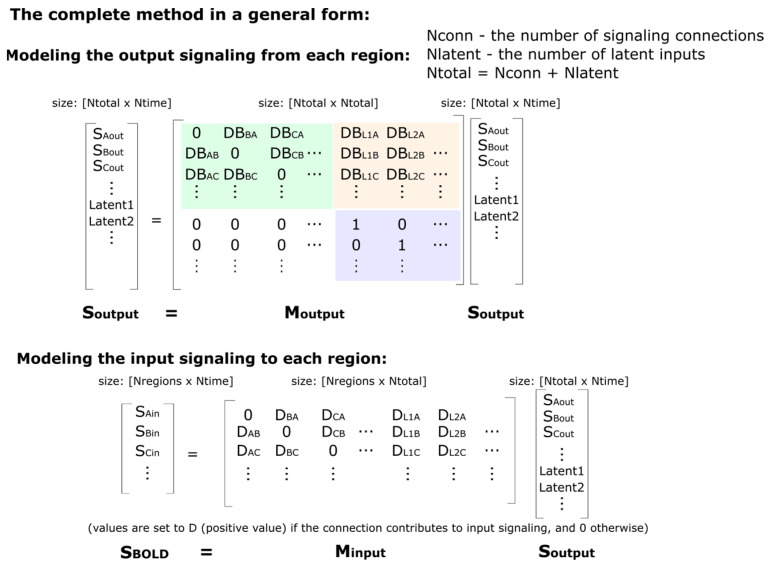
A more general description of the SAPM method. The green highlighted portion of the M_output_ matrix indicates the connections between regions, whereas the light-orange highlighted portion indicates the influence of the latent inputs on the output signals. The purple highlighted portion is an identity matrix that corresponds with the latent inputs, and this portion ensures that the matrix will have an eigenvalue equal to 1 for each latent input in the model.

**Figure 7 brainsci-13-01568-f007:**
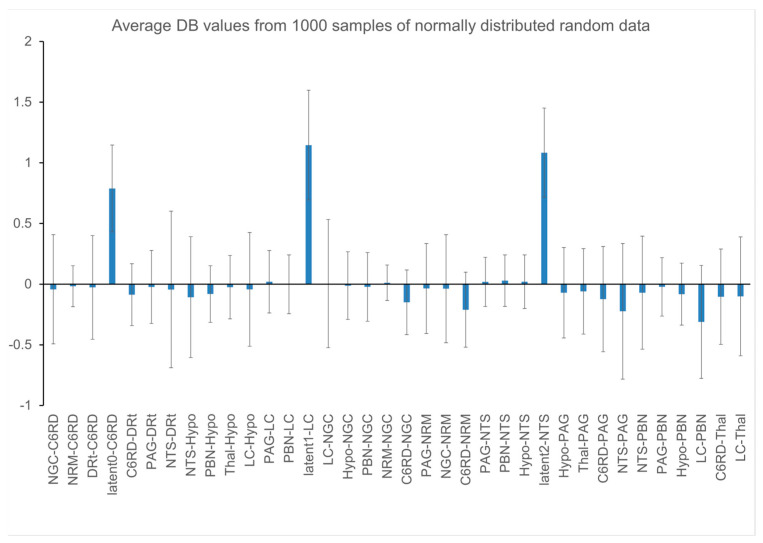
Plot of DB value distributions obtained with 1000 simulations with null data.

**Figure 8 brainsci-13-01568-f008:**
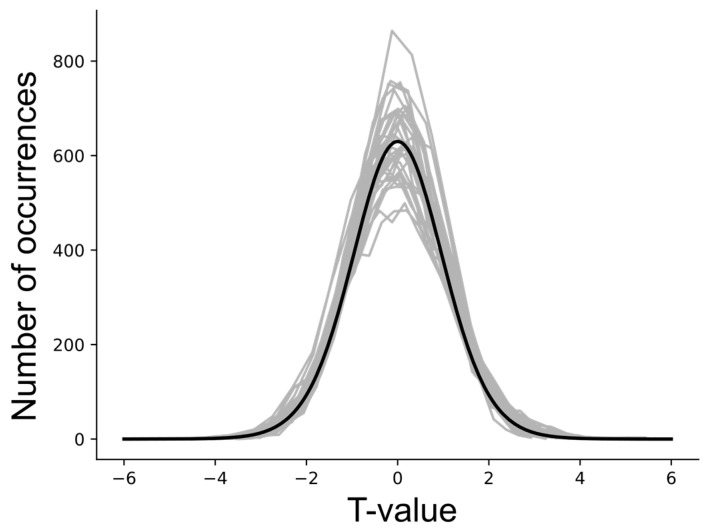
T-value distributions obtained with null data for each connection in the network, with 5000 simulated groups of data involving 20 participants per group. The gray lines indicate the distributions for each connectivity value. The black line shows a corresponding Student’s T-distribution for 20 samples.

**Figure 9 brainsci-13-01568-f009:**
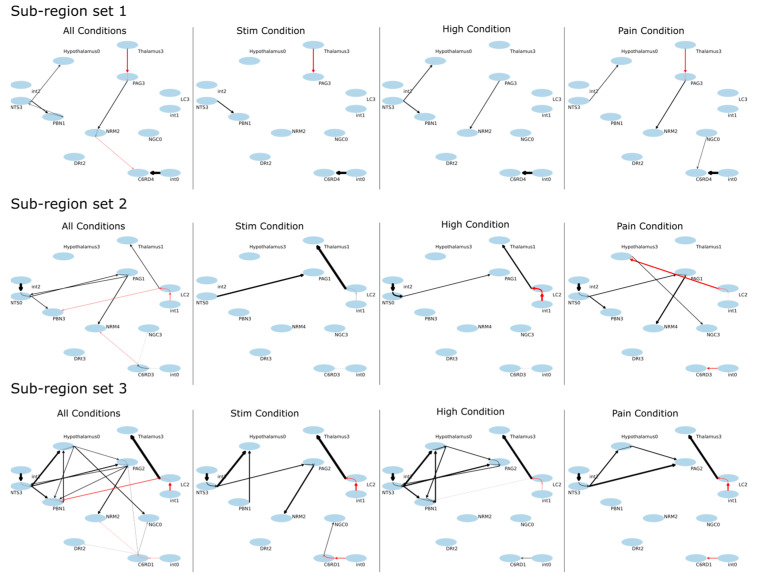
Results of SAPM analysis applied to all of the data sets (All Conditions), as well as to each of the study conditions, separately (“Stim”, “High”, “Pain”). Results are shown in the form of connectivity plots between regions. The ovals represent the anatomical sub-regions used in the analysis, with labels showing the abbreviated name and the sub-region number. Latent inputs are labeled “int0”, “int1”, and “int2”, referring to “intrinsic” input. Results are shown for three different selections of combinations of sub-regions. The arrows represent the direction of signaling, with black lines indicating excitatory effects (greater input signaling results in greater output signaling) and red lines indicating inhibitory effects (greater input signaling results in lower output signaling).

**Figure 10 brainsci-13-01568-f010:**
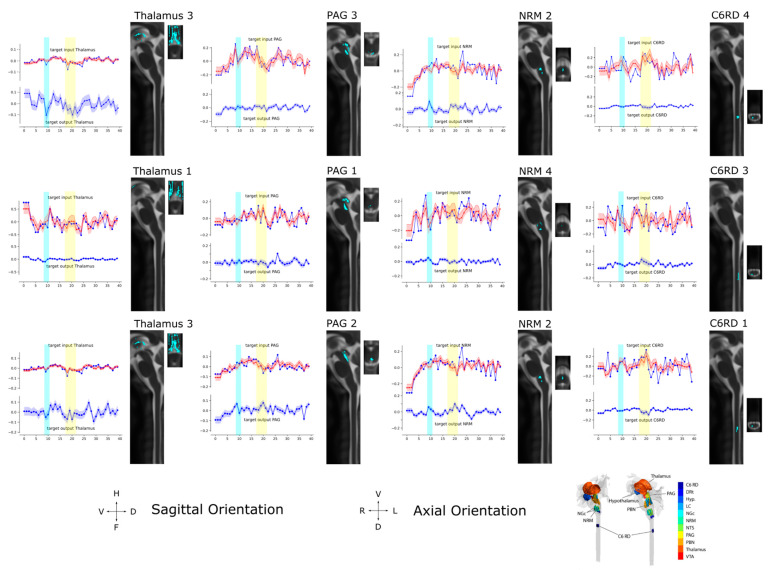
Input/Output time courses and anatomical regions for the thalamus, PAG, NRM, and C6RD region of the spinal cord averaged over 55 participants in all three study conditions (All Conditions). The results correspond to the sets of sub-regions described in Figure 9. The time-series responses plotted in red in the upper half of each plot show the measured BOLD responses. The blue lines show the modeled time-series responses for the input signaling, overlying the measured BOLD responses in the upper half of each plot, and the output signaling is shown in the lower half of each plot. The time period when participants were informed of the type of stimulus to expect is indicated with a blue bar and the time period when the stimulus was applied is indicated with a yellow bar. The corresponding anatomical extent of each sub-region is shown in sagittal and axial views, with the orientation indicated at the bottom of the figure. A 3D representation of the locations of the anatomical regions is also shown as a reference in the lower right corner of the figure.

**Table 1 brainsci-13-01568-t001:** Participant characteristics and pain ratings for each study/condition.

Study Group	N (F:M)	Average Age	Pain Rating	Temp. (°C)	Stimulus	Paradigm (Pre–Stim–Post)	Conditions
Pain Stimulation	16 (13:3)	21.3 ± 2.3	51.9 ± 8.6 in the “Stim” condition, no ratings for the “No Stim” condition	48.0 ± 0.8	10 heat contacts, every 3 s	120 s–30 s–120 s	10 repeated runs; stimulation runs interleaved with runs without stimulation
Two Pain	20 (10:10)	22.8 ± 3.0	48.8 ± 10.1 (High), 42.7 ± 11.5 (Low)	50.5 ± 0.8	10 heat contacts, every 3 s	120 s–30 s–120 s	10 repeated runs; runs with a calibrated temperature interleaved with runs that participants believed were at a lower temperature
Touch Pain	19 (9:10)	24.4 ± 7.0	44.7 ± 11.5 (Pain), 9.9 ± 3.8 (Sensation)	51.1 ± 1.3 (Pain), 40.0 (Sensation)	10 heat contacts, every 3 s	120 s–30 s–120 s	10 repeated runs; noxious stimulation runs interleaved with low temperature (sensation) runs
	57 total, 55 in high-pain conditions					270 s total	

**Table 2 brainsci-13-01568-t002:** Details of SAPM results obtained with sub-region set 1. Values in bold-face font are significantly different than zero (p_corr_ < 0.05) after family-wise error rate correction for multiple comparisons. Corresponding values that did not reach significance in other conditions are listed as well, for comparison.

Connection	All Conditions		Stim		High		Pain		Ref Value
	DB	T	DB	T	DB	T	DB	T	
**NTS–PBN**	**0.273 ± 0.053**	**6.43**	**0.341 ± 0.062**	**6.58**	**0.300 ± 0.095**	**3.91**	0.162 ± 0.101	2.30	−0.070
**PAG–NRM**	**0.254 ± 0.049**	**5.92**	0.230 ± 0.086	3.09	**0.241 ± 0.074**	**3.73**	**0.323 ± 0.093**	**3.85**	−0.036
**Thal–PAG**	**−0.356 ± 0.052**	**5.71**	**−0.393 ± 0.079**	**−4.23**	−0.259 ± 0.072	−2.79	**−0.326 ± 0.077**	**−3.45**	−0.059
**NTS–Hypo**	**0.167 ± 0.053**	**5.18**	0.084 ± 0.105	1.82	**0.263 ± 0.088**	**4.19**	**0.225 ± 0.089**	**3.75**	−0.108
**NRM–C6RD**	**−0.084 ± 0.019**	**3.55**	−0.016 ± 0.025	0.02	−0.114 ± 0.033	−2.92	−0.053 ± 0.034	−1.06	−0.017
**PBN–NTS**	**0.112 ± 0.025**	**3.30**	0.109 ± 0.035	2.28	0.016 ± 0.043	−0.30	0.118 ± 0.039	2.25	0.029
**NGC–C6RD**	0.018 ± 0.064	0.94	0.021 ± 0.046	1.41	−0.091 ± 0.135	−0.36	**0.151 ± 0.056**	**3.43**	−0.043

**Table 3 brainsci-13-01568-t003:** Goodness-of-fit (R^2^) values for each study condition and set of sub-regions, in comparison with R^2^ values obtained with 1000 simulated sets of null data.

Condition	Samples	R^2^ Average		R^2^ Total	
		MEAN ± STD	Range	Mean ± Std	Range
**Null Data**	**1000**	**0.255 ± 0.010**	**0.222 to 0.296**	**0.255 ± 0.011**	**0.214 to 0.294**
**Sub-region set 1**					
All Pain	55	0.351 ± 0.024	0.288 to 0.398	0.337 ± 0.077	0.207 to 0.600
Stim	16	0.343 ± 0.025	0.289 to 0.387	0.319 ± 0.069	0.212 to 0.480
High	20	0.338 ± 0.020	0.305 to 0.382	0.316 ± 0.051	0.237 to 0.425
Pain	19	0.369 ± 0.020	0.332 to 0.400	0.383 ± 0.081	0.306 to 0.606
**Sub-region set 2**					
All Pain	55	0.342 ± 0.025	0.293 to 0.398	0.375 ± 0.071	0.201 to 0.521
Stim	16	0.339 ± 0.028	0.293 to 0.398	0.380 ± 0.077	0.197 to 0.486
High	29	0.337 ± 0.024	0.294 to 0.396	0.392 ± 0.062	0.278 to 0.519
Pain	19	0.348 ± 0.021	0.311 to 0.401	0.377 ± 0.089	0.196 to 0.496
**Sub-region set 3**					
All Pain	55	0.371 ± 0.032	0.312 to 0.522	0.290 ± 0.072	0.142 to 0.536
Stim	16	0.366 ± 0.024	0.316 to 0.406	0.265 ± 0.056	0.153 to 0.351
High	29	0.382 ± 0.040	0.309 to 0.513	0.292 ± 0.075	0.180 to 0.525
Pain	19	0.363 ± 0.022	0.321 to 0.402	0.308 ± 0.071	0.149 to 0.485

## Data Availability

The data used in this study were from previous studies and are available upon reasonable request from the corresponding author. The data are not publicly available as of yet. The analysis software, Pantheon, is freely available on GitHub at https://github.com/stromanp/pantheon-fMRI (accessed on 7 November 2023).

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
