# Peer review of "Structural and Physiological Modeling (SAPM) for the Analysis of Functional MRI Data Applied to a Study of Human Nociceptive Processing"

_brainsci, 2023, doi:10.3390/brainsci13111568_

Round 1

Reviewer 1 Report

Comments and Suggestions for Authors

The authors developed a novel method, called Structural and Physiological Modeling (SAPM) for analyzing connectivity between brain regions using functional magnetic resonance imaging (fMRI) data. The analysis shows that SAPM can provide reliable and detailed information about neuroanatomy and signaling from brain regions, enhancing our understanding of fMRI data beyond what was previously possible. This approach is well-crafted. Great work! However, prior to recommending manuscript acceptance to the editor, please address the following questions to further refine your work.

1. Can you provide the link to "This software, “Pantheon”, is freely 140 available on GitHub."?   2. Line 144: The authors mentioned, "The anatomical template and corresponding maps of anatomical regions have been extensively described in previous works [3, 8, 11, 12]." Line 154: The authors mentioned, "The anatomical regions to be used for subsequent analyses were identified using the region maps described above." However, it remains unclear how these maps were actually identified. Including a brief explanation of the identification process would improve the clarity of the methodology, which is currently lacking in the existing version.

Author Response

We thank the reviewer for their helpful comments.  We have addressed each point below, including revisions to the manuscript.

For clarity, the reviewer’s comments/questions are repeated here in italics, followed by our responses.

  1. Can you provide the link to "This software, “Pantheon”, is freely 140 available on GitHub."?

The link to GitHub is provided in the data availability statement and we have now also added it to the methods section.

  1. Line 144: The authors mentioned, "The anatomical template and corresponding maps of anatomical regions have been extensively described in previous works [3, 8, 11, 12]." Line 154: The authors mentioned, "The anatomical regions to be used for subsequent analyses were identified using the region maps described above." However, it remains unclear how these maps were actually identified. Including a brief explanation of the identification process would improve the clarity of the methodology, which is currently lacking in the existing version.

We have added the following text to describe how the region maps were identified:

“The anatomical reference image (template) was defined spanning across the brain and spinal cord regions by combining the MNI152 template and PAM50 template as described by De Leener et al. (De Leener et al. 2018). Corresponding anatomical region-of-interest maps were defined from multiple sources including the CONN15e software package, freely shared anatomical maps, and anatomical descriptions (Lang and Bartram 1982, Talairach and Tournoux 1988, Millan 2002, Keren et al. 2009, Naidich et al. 2009, Whitfield-Gabrieli and Nieto-Castanon 2012, Leijnse and D'Herde 2016, De Leener et al. 2017, Pauli et al. 2018, Chiang et al. 2019, Liebe et al. 2020, Stroman et al. 2020).  These sources were combined to create a single anatomical map.”

Reviewer 2 Report

Comments and Suggestions for Authors

This study sought to validate SAPM, a method that extends basic structural equation modelling (SEM) for the analysis of fMRI data, recently published by the authors in another journal. It is clear that a lot of work has gone into this study and the manuscript is clearly presented. However, I think the authors have gone in the wrong direction by deploying SEM for fMRI data. Additionally, the validation presented here doesn’t quite take the right approach. I’ll expand on these points as follows:

- While the authors have done impressive work in extending SEM to capture inputs and outputs from each brain region, I don’t think SEM is the right approach to take in principle. In case the authors are not aware, SEM fell out of favour in fMRI analysis in the mid-2000s, due to certain shortcomings. In particular, the challenges with handling loops, a failure to separate neural and haemodynamic contributions to the fMRI data and the difficulty of comparing non-nested models. These issues are now typically addressed by using state-space models, in conjunction with Bayesian model inversion methods, which together are referred to as Dynamic Causal Modelling (DCM) for fMRI. Formally, SEM is a special case of DCM for fMRI, where the neural activity is directly observed rather than being mediated by hemodynamics, and where the within-region intrinsic dynamics are fixed to uniform values rather than being estimated from the data on a per-region basis. For a detailed comparison of SEM and DCM, please see Penny et al., 2004, NeuroImage - https://doi.org/10.1016/j.neuroimage.2004.07.041 . The extension to SEM validated by the authors does not, as far as I can see, overcome the disadvantages of SEM relative to DCM that are already established.

- The simulated null data used for the validation in this paper were normally distributed values (i.e., only noise). A more biologically realistic test would be to simulate fMRI data under different hypothesised neural network architectures for the pain system, and identify whether SAPM can correctly distinguish them. These simulated fMRI timeseries would need to be generated by a model equipped to translate connectivity among neural populations to haemodynamics, i.e. a DCM. However, in practice, I wouldn’t recommend pursuing this – I think the Penny et al. (2004) study already captures the main ideas for why SEM is not the right way to proceed.

I am sorry to be negative and hope that this feedback will be useful for the authors.

Author Response

We appreciate the reviewer's feedback and would like to clarify the primary objective of our paper. Our main focus is to demonstrate the SAPM method rather than the SEM method, and we do not intend to make a direct comparison between either of these methods and DCM. While we acknowledge the reviewer’s point, SEM continues to be used for fMRI data analysis as demonstrated by a relatively recent review article on the topic [1].

In order to point out that our approach differs from the original SEM method as described for economics, and to acknowledge potential weaknesses that have previously been identified (as suggested by the reviewer), we have added the following text to section 2.6 where we describe the SAPM method:

“. SAPM is an extension of our SEM method because the direction of interactions is inferred based on anatomical information and the fitting method is based on linear regression between time-series responses. This differs from the original SEM method and later variations that have been applied to neuroimaging data which are based on modeling the covariance between regions [2-4].”

However, we respectfully disagree with the reviewer’s claim that SEM is not the right way to proceed, based on the fact that our SEM approach is not the same as the approach described by Penny et al. (2004). While the current paper presents an extension of SEM in the form of SAPM, we have demonstrated the use of our SEM approach in ten different published studies over about eight years [5-14]. We have demonstrated that this method can provide reproducible results and demonstrate important features of BOLD fMRI responses. Furthermore, we have demonstrated consistent connectivity values between specific regions as well as values that vary in relation to pain ratings, or that differ between study conditions or with different study groups.

The SEM approach that we have demonstrated is simpler than the approach described in Penny et al. (2004). The approach described in Penny et al. is not a stable method of determining the connectivity parameters for SEM because it relies on modeling the covariances between BOLD responses. This is as described in the original paper by McArdle and McDonald (1984) for data in the field of economics rather than for neuroimaging. The SEM method that we have employed is essentially a series of linear regressions, one for each “target” region with multiple “source” regions. The SAPM method models the entire network simultaneously, and is more similar to the original method described by McIntosh and Gonzalez-Lima in that regard, but the fitting of the data to the model is still based on the time-series responses and not the covariance between them. Again, the method is based primarily on linear regression as opposed to optimizing the match to the covariance of the data. Moreover, our SEM and SAPM methods allow for reciprocal connections or “loops” because we are not using the same approach to determine the connectivity parameters. The approach used in Economics, and as described by Penny et al., also requires a unique solution. Our approach does not require a unique solution, it requires only a solution that provides the best-fit of our data to the network model in order to provide a plausible explanation for the observed BOLD responses. The possible solutions are constrained to the one that provides the best fit with the smallest connectivity values.

Dynamic causal modeling relies on temporal relationships between BOLD signals and requires high temporal resolution and is a separate topic. The SEM and SAPM methods that we discuss in the present paper are not similar approaches and do not rely on the timing of hemodynamics or trying to capture differences in timing. Our methods look at the consistent features of relationships between states of the BOLD signals across a network of regions on slower time-scales. Any inferences about the direction of relationships between regions are based on the known neuroanatomy.

We note that in the Penny et al. paper the authors point out: “We have described SEM as implemented in the majority of applications to functional brain imaging data. … In the wider SEM world, however, SEMs vary in their generative models, estimation procedures and styles of inference. It is possible, for example, to define SEMs with exogenous variables.” Our SAPM approach does indeed use exogenous variables, as was described by McArdle and McDonald (1984). We believe that our continued development of SEM and SAPM methods has generated important results and valuable methods.

Finally, we believe that the applications of SEM and SAPM that we have already demonstrated provide strong evidence for the validity of the methods. Arguments against these methods appear to be based only on potential problems that are suggested by older mathematical approaches and assumptions and constraints on the model that are not applied with our methods.

In response to the final point raised by the reviewer, the purpose of null tests is to identify the distributions of fit parameters that could occur by random chance, for the purposes of statistical testing. Modeling “biologically realistic” data would present a whole new set of problems and assumptions that would need to be validated, and is a different topic. We have already demonstrated the reproducibility and sensitivity of the results obtained with SAPM by testing it with real data.

References

  1. Guardia-Olmos, J., M. Pero-Cebollero and E. Gudayol-Ferre. "Meta-analysis of the structural equation models' parameters for the estimation of brain connectivity with fmri." Front Behav Neurosci 12 (2018): 19. 10.3389/fnbeh.2018.00019. https://www.ncbi.nlm.nih.gov/pubmed/29497368.
  2. McArdle, J. J. and R. P. McDonald. "Some algebraic properties of the reticular action model for moment structures." Br J Math Stat Psychol 37 ( Pt 2) (1984): 234-51. http://www.ncbi.nlm.nih.gov/pubmed/6509005.
  3. McIntosh, A. R. and F. Gonzalez-Lima. "Structural modeling of functional neural pathways mapped with 2-deoxyglucose: Effects of acoustic startle habituation on the auditory system." Brain Res 547 (1991): 295-302. http://www.ncbi.nlm.nih.gov/pubmed/1884204.
  4. Penny, W. D., K. E. Stephan, A. Mechelli and K. J. Friston. "Modelling functional integration: A comparison of structural equation and dynamic causal models." Neuroimage 23 Suppl 1 (2004): S264-74. S1053-8119(04)00389-1 [pii] 10.1016/j.neuroimage.2004.07.041. http://www.ncbi.nlm.nih.gov/pubmed/15501096.
  1. Ioachim, G., J. M. Powers, H. J. M. Warren and P. W. Stroman. "Coordinated human brainstem and spinal cord networks during the expectation of pain have elements unique from resting-state effects." Brain Sci 10 (2020): 10.3390/brainsci10090568. https://www.ncbi.nlm.nih.gov/pubmed/32824896.
  2. Khan, H. S. and P. W. Stroman. "Inter-individual differences in pain processing investigated by functional magnetic resonance imaging of the brainstem and spinal cord." Neuroscience 307 (2015): 231-41. 10.1016/j.neuroscience.2015.08.059. http://www.ncbi.nlm.nih.gov/pubmed/26335379.
  3. Stroman, P. W. "Validation of structural equation modeling methods for functional mri data acquired in the human brainstem and spinal cord." Crit Rev Biomed Eng 44 (2016): 227-41. 10.1615/CritRevBiomedEng.2017020438. http://www.ncbi.nlm.nih.gov/pubmed/29199575.
  4. Powers, J. M., G. Ioachim and P. W. Stroman. "Evidence for integration of cognitive, affective, and autonomic influences during the experience of acute pain in healthy human volunteers." Front Neurosci 16 (2022): 884093. 10.3389/fnins.2022.884093. https://www.ncbi.nlm.nih.gov/pubmed/35692431.
  5. Stroman, P. W., R. L. Bosma, A. I. Cotoi, R. H. Leung, J. Kornelsen, J. M. Lawrence-Dewar, C. F. Pukall and R. Staud. "Continuous descending modulation of the spinal cord revealed by functional mri." PLoS One 11 (2016): e0167317. 10.1371/journal.pone.0167317. http://www.ncbi.nlm.nih.gov/pubmed/27907094.
  6. Stroman, P. W., G. Ioachim, J. M. Powers, R. Staud and C. Pukall. "Pain processing in the human brainstem and spinal cord before, during, and after the application of noxious heat stimuli." Pain 159 (2018): 2012-20. 10.1097/j.pain.0000000000001302. https://www.ncbi.nlm.nih.gov/pubmed/29905656.
  7. Stroman, P. W., J. M. Powers, G. Ioachim, H. J. M. Warren and K. McNeil. "Investigation of the neural basis of expectation-based analgesia in the human brainstem and spinal cord by means of functional magnetic resonance imaging." Neurobiol Pain 10 (2021): 100068. 10.1016/j.ynpai.2021.100068. https://www.ncbi.nlm.nih.gov/pubmed/34381928.
  8. Stroman, P. W., H. J. M. Warren, G. Ioachim, J. M. Powers and K. McNeil. "A comparison of the effectiveness of functional mri analysis methods for pain research: The new normal." PLoS One 15 (2020): e0243723. 10.1371/journal.pone.0243723. https://www.ncbi.nlm.nih.gov/pubmed/33315886.
  9. Warren, H. J. M., G. Ioachim, J. M. Powers and P. W. Stroman. "How fmri analysis using structural equation modeling techniques can improve our understanding of pain processing in fibromyalgia." J Pain Res 14 (2021): 381-98. 10.2147/JPR.S290795. https://www.ncbi.nlm.nih.gov/pubmed/33603453.
  10. Yessick, L. R., C. F. Pukall, S. F. Chamberlain and P. W. Stroman. " An investigation of descending pain modulation in women with provoked vestibulodynia (pvd): Alterations of spinal cord and brainstem connectivity." Frontiers in Pain Research 12 August 2021 (2021): 10.3389/fpain.2021.682483.

Reviewer 3 Report

Comments and Suggestions for Authors

The manuscript titled "Structural and Physiological Modeling (SAPM) for the analysis of functional MRI data applied to a study of human nociceptive processing" proposed a novel approach for connectivity analysis based on fMRI data for deep structures involving the  brainstem and spinal cord. 

There are several strengths to recommend. SAPM offers a novel and comprehensive approach to analyzing connectivity in fMRI data by combining information about blood oxygenation, anatomy, and physiology. SAPM's ability to provide information about input and output signaling from anatomical regions is a significant strength. It helps researchers understand how different brain regions communicate and whether signaling is predominantly inhibitory or excitatory. The method allows for the exploration of mechanistic insights into brain function, which can be valuable for understanding the neural underpinnings of complex cognitive processes and disorders. The study's emphasis on validation, reproducibility, and statistical thresholds adds rigor to the findings, enhancing the confidence in the results. In addition, the application of SAPM to investigate human nociceptive processing in the brainstem and spinal cord demonstrates its practical utility in studying real-world phenomena and clinical applications.

There are also several areas that need further consideration and clarifications. 

1. The use of a two-source model in SAPM may be overly simplified for modeling complex brain connectivity patterns. This simplification can potentially limit the accuracy of connectivity representations.

2. The reliance on BOLD signals in fMRI data introduces temporal and spatial discrepancies compared to direct neural signals. This can complicate the precise modeling of excitatory and inhibitory interactions.

3. SAPM, like many neuroimaging computational modelling, may struggle to account for the significant inter-individual variability in brain connectivity, potentially leading to limitations in generalizability. 

4. While SAPM can model connectivity, it's essential to acknowledge that it doesn't establish causal relationships between brain regions. Determining causality remains a challenging aspect of neuroimaging research.

5. The research team could explore more complex connectivity models that go beyond the two-source model. Consideration of higher-order interactions and indirect pathways may provide a more accurate representation of brain connectivity.

6. There are other connectivity analysis models, such as dynamic causal modeling (DCM) and Granger Causality Analysis, may help researchers infer causal relationships among brain regions, going beyond correlational findings. Comparing the different approaches in the analysis of the dataset can be helpful to understand the relative advantages and limitations of the models and serve as additional validation for the connectivity models, their statistical significance and structural complexity. For instance, the SAPM model does not account for some potential mediators or unmeasured variables that could influence the connections between regions. As a result, some relationships may appear to exist or be misrepresented. How directionality is specifically taken into account needs consider anatomical basis and possible information flow with multiple sources that carry different strengths of influence. 

7. How the input and output matrixes are differentially modulated by the input type and stimulus strength needs to be taken into account. There may be direct and indirect connections among the target regions. Computationally modelling the indirect connections that contribute to excitatory or inhibitory pathways could be challenging. Moreover, interpreting the results of complex connectivity analyses, especially when involving indirect connections or higher-order interactions, can be more challenging. Careful consideration of the biological plausibility of the findings is crucial.

8. Discussion can go beyond the current approach and data set for the modelling. Future work could consider multimodal integration, biophysical modelling, experimental validation including, for instance, other stimulation protocols and clinical vs. neurotypical populations. 

Author Response

We thank the reviewer for their helpful comments.  We have addressed each point below, including revisions to the manuscript.

For clarity, the reviewer’s comments/questions are repeated here in italics, followed by our responses.

  1. The use of a two-source model in SAPM may be overly simplified for modeling complex brain connectivity patterns. This simplification can potentially limit the accuracy of connectivity representations.

The two-source model was used only with the SEM method in the process of identifying potential connections for the purposes of validating the choice of network model.  The SAPM method is not a two-source method, the relationships between regions are modeled simultaneously for all regions/connections in the model. We have attempted to make this more clear in the text with the following addition to section 2.5.1:

“Note that this “two-source” SEM method was used only for validating the network model, whereas the full network model is used for the SAPM method.”

  1. The reliance on BOLD signals in fMRI data introduces temporal and spatial discrepancies compared to direct neural signals. This can complicate the precise modeling of excitatory and inhibitory interactions.

We fully agree with the reviewer’s point and have made every effort throughout the manuscript to try to be clear that the results demonstrate “inhibitory effects” or “excitatory effects” and that the connectivity values reflect the “apparent transmission effect”, referring to whether the effect appears to be excitatory or inhibitory.  It is also well-known that the temporal resolution of fMRI data is very low compared to direct neural signals and the data show only average effects over many neurons and neural signaling events. Nonetheless, BOLD signal variations have been shown to be related to input signaling and the metabolic demand of the tissues, and therefore provide information about neural activity over a region. We are careful in our interpretations of the results to acknowledge the spatial and temporal limitations of the available information.

  1. SAPM, like many neuroimaging computational modelling, may struggle to account for the significant inter-individual variability in brain connectivity, potentially leading to limitations in generalizability.

A key point of our SAPM method is that we applied it to data from each participant. This fact is mentioned multiple times in the Methods section. As a result, we are able to demonstrate individual differences with our results.  Our first paper on this method was titled “Proof-of-concept of a novel structural equation modelling approach for the analysis of functional MRI data applied to investigate individual differences in human pain responses”. We believe that for the purposes of studying nociceptive responses that individual differences are extremely important, so the reviewer’s point is well-taken.

  1. While SAPM can model connectivity, it's essential to acknowledge that it doesn't establish causal relationships between brain regions. Determining causality remains a challenging aspect of neuroimaging research.

We agree that this is a very important point. Throughout the manuscript we attempted to be clear with our choice of wording that the results reflect relationships between temporal signal variations between regions. The method we have demonstrated does not use only fMRI data, but also includes known anatomical connections that demonstrate the direction of signaling, and multiple simultaneous connections that are different for each region, and relationships between BOLD signal variations and the underlying physiological processes. This is the key strength of the SAPM method, that more information is used to create a more detailed model. As a result, the model from one region to another is very different from the model of the signaling in the reverse direction. The direction of the interaction can indeed be inferred, unless the anatomical model is chosen to be overly simplistic. We cannot say that the SAPM method will always produce accurate results and demonstrate relationships that are true, because we cannot know how people will choose to apply this method in the future. However, in the present paper we believe that we have demonstrated the reliability and reproducibility of the results and that they correspond well with the known neuroanatomy.

  1. The research team could explore more complex connectivity models that go beyond the two-source model. Consideration of higher-order interactions and indirect pathways may provide a more accurate representation of brain connectivity.

We mentioned in our response to a previous comment that the two-source model was only used with SEM for validating the network model.  The SAPM method is far more complex. The study of different models, in relation to different function, etc., will hopefully be a large area of active research in the future. In the present manuscript we have discussed the limitations of interpreting the connections and the possibility of indirect pathways. We have also added to the Conclusions to acknowledge this point as follows:

“The results demonstrate input and output signaling that can explain observed BOLD signal variations. However, the results depend on the choice of network model and may not include all connections or all regions that contribute to neural signaling.”

  1. There are other connectivity analysis models, such as dynamic causal modeling (DCM) and Granger Causality Analysis, may help researchers infer causal relationships among brain regions, going beyond correlational findings. Comparing the different approaches in the analysis of the dataset can be helpful to understand the relative advantages and limitations of the models and serve as additional validation for the connectivity models, their statistical significance and structural complexity. For instance, the SAPM model does not account for some potential mediators or unmeasured variables that could influence the connections between regions. As a result, some relationships may appear to exist or be misrepresented. How directionality is specifically taken into account needs consider anatomical basis and possible information flow with multiple sources that carry different strengths of influence.

We believe that this is addressed in responses to previous points raised by the reviewer. We have discussed in the manuscript some of the limitations and the possibility of connections being mediated through other sources that are not included in the model. We feel that this is already a long and complex manuscript.  Adding a comparison of the SAPM method to other methods such as DCM and Granger Causality would divert the focus of the manuscript from its main point and would reduce its quality. Each of these methods has a purpose that depends on the research question that is being asked. They may not even be comparable.  That can be a topic for a future paper.

  1. How the input and output matrixes are differentially modulated by the input type and stimulus strength needs to be taken into account. There may be direct and indirect connections among the target regions. Computationally modelling the indirect connections that contribute to excitatory or inhibitory pathways could be challenging. Moreover, interpreting the results of complex connectivity analyses, especially when involving indirect connections or higher-order interactions, can be more challenging. Careful consideration of the biological plausibility of the findings is crucial.

We feel that this is essentially the same point as the previous one. We provide quite a thorough discussion of how the SAPM results compare with known neuroanatomy and neurophysiology. We believe that we have demonstrated careful consideration of the biological plausibility of the findings.

  1. Discussion can go beyond the current approach and data set for the modelling. Future work could consider multimodal integration, biophysical modelling, experimental validation including, for instance, other stimulation protocols and clinical vs. neurotypical populations.

Yes, these points will be excellent topics for future studies. This present manuscript is essential for this present stage of demonstrating the method before we go on to apply it to the study of clinical populations or combining the results with other modalities. This is the start of a new method. This second paper ever written about SAPM is not expected to answer every question about it. We believe that we have done quite a thorough job however of validating the statistical methods, demonstrating the reliability of the results, and providing a demonstration of how SAPM can be used.

Round 2

Reviewer 2 Report

Comments and Suggestions for Authors

Following the first round of peer review, the authors’ response included helpful clarification on the differences between SEM and SAPM, as well as a helpful summary of their previous work, and they may wish to consider incorporating more of this response into the manuscript. There are, as I see it, two considerations – 1) whether the newly introduced method, SAPM, is fit for purpose in principle, and 2) the specifics of this manuscript, which assesses its validity using empirical data.

Starting with the second consideration – I think the validation presented here has been conducted well and is clearly reported. The choice of nodes in the network was well-motivated, enabling strong hypotheses, which the authors were able to discuss in relation to the connectivity estimated by SAPM. The use of three datasets was a nice decision, enabling the consistency across datasets to be evaluated. However, as I will revisit, this really needs to be paired with validation using simulated data, because the ground truth is not known.

My concern about SAPM stems from first principles, which I referenced in the first round of review and I will expand on here, before suggesting a possible way forward.

First, I will state my understanding about the biological system being studied, before comparing it to how SAPM models the data. As far as I understand, neural populations in different regions of the brain communicate via long range connections, establishing fast dynamics at the synaptic level. Under a mean-field assumption, the input to a target brain region can be summarised reasonably well by the mean activity of neural activity in source regions - licensing the use of one variable per brain region summarising that region’s activity. Local neural activity in each region then undergoes a non-linear transform (via neurovascular coupling, vascular dynamics and magnetic effects) to generate local BOLD responses. Because the BOLD response acts like a low-pass filter, with fMRI we are only sensitive to slow population-level neural activity that evolves over a few seconds, which gives rise to even slower vascular dynamics that takes about 30 seconds for a full duty cycle.

Now, to contrast this process-model against the statistical model in SAPM. My understanding is that SAPM models the “output” of brain regions as a weighted linear mixture of the outputs of other regions (plus latent unmodelled noise). The “inputs” of brain regions are then obtained as a linear mixture of the outputs of connected regions. My concern is that SAPM lacks a hemodynamic model, and treats BOLD responses as being coupled across regions, which they are not (the same concerns apply to SEM when applied to fMRI). In detail:

-          In SAPM there is no separation of timescales for inputs and outputs - it treats them as the same. That’s not what happens biologically – neural inputs have a faster timescale than the BOLD responses that forms the output. Thus, in SAPM, an input can only happen when other regions have already expressed their BOLD responses, many seconds later than the actual input. I am therefore unclear how to interpret the input parameters in SAPM.

-          The lack of a haemodynamic model means that overlapping BOLD responses cannot be deconvolved using SAPM. Thus, SAPM can only be applied to paradigms with stimuli that are very widely spaced in time, as in the example dataset presented by the authors. Statistically efficient event-related designs are therefore precluded. If have I have understood this limitation correctly, it should be explicitly stated in the paper.

-          Non-linearities in the BOLD response, which are very well understood, cannot be captured in SAPM, as far as I can see. E.g., we would expect two stimuli that are close together in time to have a super-additive effect on the BOLD response. This is handled in standard mass-univariate GLM analysis using convolution with a first-order linear response function (canonical HRF), and in DCM analysis using a full biophysical model, but there is no equivalent here as far as I can see. Thus my expectation is that changing the interval between stimuli would change the parameter estimates in SAPM in a non-trivial manner, and I wonder if the authors have an intutition on this.

-          Rather than assume linear relationships between brain regions, which in turn gives rise non-linearly to BOLD responses, SAPM assumes a linear relationship between BOLD responses in different brain regions. I wasn’t clear what the justification for this assumption was.

-          Different brain regions have different haemodynamic delays (more precisely, they take different times to reach their peak). How are these differences reflected in SAPM parameters? Would a difference in haemodynamic response time change the connectivity parameters? I expect that it would do, because there are no terms in the model to capture these variable response times. This issue is handled in convolution GLM models through the inclusion of spatial and dispersion derivative terms (i.e., a suitable set of basis functions), and in DCM via the region-specific parameterisation of the haemodynamic model.

Note that none of these issues can be demonstrated using empirical fMRI data, where the ground truth is not known, and therefore they are not revealed in the validation presented here or in the original SAPM paper. Hence, my previous suggestion of using simulated data to investigate how SAPM performs in different scenarios.

As well as being upfront about these issues in the manuscript, the question that I think the authors need to demonstrate is whether SAPM offers benefits that outweigh these limitations – and does it offer advantages over established methods for fMRI analysis. I don’t think this case has been made. A way to overcome some of these issues would be to combine the linear model used in SAPM with a canonical haemodynamic response function, thereby implicitly changing the interpretation of the connectivity parameters so that they relate to the neural, rather than haemodynamic level. This would also provide a faster method for estimating connectivity in large-scale models than is possible with DCM, enabling a more exploratory approach. However, it would need to be demonstrated that this would offer advantages over the recently introduced Regression DCM, which represents the state of the art (e.g., Frassle et al., 2017; Frassle et al. ,2021).

Finally, I will just address a misunderstanding in the authors’ response to my previous comments relating to DCM for fMRI, in case it’s helpful for contextualising SAPM against previous work:

“Dynamic causal modeling relies on temporal relationships between BOLD signals and requires high temporal resolution and is a separate topic.”

I thought it would be helpful for the authors to consider the relationship between DCM and SAPM because these methods employ the same form of connectivity model, except in DCM it is deployed as a model of neural connectivity, and in SAPM it’s deployed as a tool for quantifying BOLD connectivity. The key difference between the methods is that DCM supplements the connectivity model with a haemodynamic model. Contrary to the authors’ comment, there is almost no useful precise temporal information in fMRI and neither DCM or SAPM requires this – rather, it’s the amplitude of the signal that is the useful data feature in fMRI. DCM also does not require high temporal resolution (see the original publication which used a block design). Nevertheless, the use of a haemodynamic model in DCM does enable fast experimental designs (i.e., event-related designs), because overlapping BOLD responses can be deconvolved.

In summary, I think this paper does a good job of validating that SAPM gives reasonable results in the sense that they are consistent with the authors’ hypotheses and have consistency across datasets. However, I suggest there are more fundamental limitations with the form of the model which limit the interpretability of the results - issues that cannot not be demonstrated with the kind of validation presented here, using empirical data where the ground truth is not known. I therefore do not think a strong enough case has yet been made for deploying SAPM in practice.

Reviewer 3 Report

Comments and Suggestions for Authors

I think the responses are good. I recommend acceptance. Maybe some responses can be incorporated in the main text. For instance,  future directions can be elaborated with a couple of sentences in the proof-reading process. Thank you for your important contribution to the field. 

Author Response

We thank the reviewer for their helpful comments and the time they have taken to review our manuscript.  We have added comments on future directions to the Discussion section, as suggested. These are incorporated into the revisions made in response to other reviewers’ comments in the last paragraph of the Discussion section.